# Validation and implementation of a mobile app decision support system for prostate cancer to improve quality of tumor boards

Yasemin Ural[1]*, Thomas Elter[2], Yasemin Yilmaz[1], Michael Hallek[2], Rabi Raj Datta[3], Robert Kleinert[3], Axel Heidenreich[1], David Pfister[1]

**1** University of Cologne, Faculty of Medicine and University Hospital Cologne, Department of Urology, Uro-Oncology and robot assisted surgery, Germany, **2** University of Cologne, Faculty of Medicine and University Hospital Cologne, Department I of Internal Medicine, Center for Integrated Oncology Aachen Bonn Cologne Duesseldorf, Germany, **3** University of Cologne, Faculty of Medicine and University Hospital Cologne, Department of General, Visceral, Cancer and Transplantation Surgery, Germany

* uralya@hotmail.com

**Data Availability Statement:** Our data was published on the open repository Zenodo (DOI: 10.1371/journal.pdig.0000054.). The full data set can be accessed by everyone using the following

## Abstract

Certified Cancer Centers must present all patients in multidisciplinary tumor boards (MTB), including standard cases with well-established treatment strategies. Too many standard cases can absorb much of the available time, which can be unfavorable for the discussion of complex cases. In any case, this leads to a high quantity, but not necessarily a high quality of tumor boards. Our aim was to develop a partially algorithm-driven decision support system (DSS) for smart phones to provide evidence-based recommendations for first-line therapy of common urological cancers. To assure quality, we compared each single digital decision with recommendations of an experienced MTB and obtained the concordance.1873 prostate cancer patients presented in the MTB of the urological department of the University Hospital of Cologne from 2014 to 2018 have been evaluated. Patient characteristics included age, disease stage, Gleason Score, PSA and previous therapies. The questions addressed to MTB were again answered using DSS. All blinded pairs of answers were assessed for discrepancies by independent reviewers. Overall concordance rate was 99.1% (1856/1873). Stage specific concordance rates were 97.4% (stage I), 99.2% (stage II), 100% (stage III), and 99.2% (stage IV). Quality of concordance were independent of age and risk profile. The reliability of any DSS is the key feature before implementation in clinical routine. Although our system appears to provide this safety, we are now performing cross-validation with several clinics to further increase decision quality and avoid potential clinic bias.

## Author summary

The quality of therapeutic decisions provided in tumor boards is perhaps the most relevant criterion for optimal cancer outcome. This tool aims to provide optimal

link: https://zenodo.org/record/6951736#.
Y5rxDOzMLyj.

**Funding:** The authors received no specific funding
for this work.

**Competing interests:** I have read the journal´s
policy and the authors of this manuscript have the
following competing interests: T. E. is the founder
and chief medical officer of the company who
developed the application "EasyOncology".

recommendations, to assess the quality on a case-by-case basis and furthermore to objectively display the quality of oncological care.

## Author summary

Everyday clinicians face the difficult task to choose the optimal treatment for their cancer patients due to the emergence of newly available therapeutics and continuously altering treatment guidelines. The resulting flood of information is impossible for clinicians to keep up with. Therefore, clinicians decide as a team, in so called tumor boards, upon the best possible cancer treatment for each patient. Even though the treatment decisions recommended by tumor boards play a critical role for the long-term survival of cancer patients, their accuracy in decision-making has hardly ever been assessed. Unfortunately, current digital tools that have been developed to support clinicians on the process of decision-making, have difficulties to provide treatment recommendations with sufficient accuracy. Therefore, we evaluated the quality of a novel decision-making application by comparing the decision concordance generated by the App with therapeutic recommendations given by a tumor board of a University Cancer Center. For newly diagnosed cancer patients we found that the novel tool matched the decisions made by the tumor board in almost 100% of the cases. These promising results not only show the potential of providing digital support for patient care, but also provide objective quality management while saving board time in favor of discussing more complex cases.

## Introduction

Uro-Oncologists and Oncologists worldwide face the challenge of ensuring that their patients receive the best possible, individualized care for their cancer disease. Keeping up with the fast developments in medicine is very difficult for many physicians, as the rapid growth of scientific knowledge leads to an almost unmanageable variety of new treatment options [1]. The large amount of data is overwhelming and consequently it becomes a demanding task, to decide for the best possible individualized therapy for the patient.

Therefore, case discussion in multidisciplinary tumor board (MTB) conferences is one of the most important factors to assure highest quality standards in oncological care. Driven by this assumption, German hospitals are required to present and discuss each cancer patient in MTB in order to get registered as certified cancer center by the German Cancer Society (Deutsche Krebshilfe) [2].

In clinical reality, certification requirements for presenting all routine cases leads to a significant increase in the number of case discussions, wasting valuable time and attention that is needed for the discussion of more complex tumor cases.

In the field of evidence-based oncology, it is almost paradoxical that the quality of individual therapeutic decisions of the tumor boards is hardly ever qualitatively assessed. Despite their widespread use in clinical routine, few data is available about the effects of tumor boards on quality of care and long-term survival of cancer patients [3,4].

This results in the need to objectively define and measure the quality of MTB at the level of individual therapeutic recommendations.

Without doubt, AI based clinical decision support systems will play a major role in the near future to close the gap between complex data and clinical decision-making [4–6].

Up to this point, systems based on artificial intelligence have been unable to offer reliable assistance in this area, as they are still failing to provide treatment recommendations with sufficient certainty even to standard questions on first-line therapy [7]. For example, one of the leading AI-based systems, Watson for Oncology, matched only 12% (stomach), 80% (colon and breast carcinoma) and 93% (ovarian carcinoma) of the treatment recommendations given by medical experts [8–13].

An important reason for the poor performance of AI systems is the lack of high-quality training datasets. Masses of normalized training data are available for AI image recognition systems, but not for AI applications that are intended to model regular oncology care.

In addition to the lack of properly organized and validated training data, another problem is the limited resource of experts initially required for human interpretation and evaluation of AI results.

At the end of the day, then, a machine learning tool can only be as good as the data available for training and the trainers who evaluate the AI results.

Another approach to provide digital therapeutic recommendations with sufficient certainty could be implemented by developing software based on clinical network expertise. This concept of expert-curated digital decision support systems (DSS) was described in Nature Biotechnology in 2018 and a comparison with approaches of artificial intelligence showed multiple benefits [14]. The main advantage described here is certainly that the expert systems seem to represent clinical reality better than the AI-based systems used so far.

The DSS smartphone application EasyOncology (EO), whose digital treatment recommendations are based on continuous matching with real tumor boards, follows this approach and led to the design of this research study.

To the best of our knowledge, tumor board decisions have not been validated on a case by case basis by using an expert-curated DSS. In addition, our study allows a direct comparison of the level of concordance reached by an AI-based DSS with an expert-curated DSS with respect to MTB decisions for prostate cancer patients.

The aim of this clinical research is to implement the aforementioned technology for validation and quality assurance of a urological tumor board at the University Hospital of Cologne.

## Materials and methods

### Study design and patients

The study obtained ethics approval by the Ethics Commission of Cologne University's Faculty of Medicine.

We present the results of 1873 cases of prostate carcinoma. We compared and analyzed the concordance rates between the tumor board recommendations of the urological multidisciplinary tumor board at the University of Cologne and the query results of the digital application "EasyOncology".

Inclusion criteria for the study were prostate cancer cases for which a therapy recommendation was given in the uro-oncological tumor board of the University Hospital in the period from 2014 to 2018. Data sets of 2412 patient cases included then and screened for exclusion criteria: 140 case discussions not addressing therapeutic procedures, such as specific questions about histopathology or how to obtain a biopsy were excluded, as well as another 264 cases without therapeutic decisions due to pending clinical information. Another 135 cases recommended for clinical trials (n = 50) or complex cases with more than one active tumor entity (n = 85) were also excluded (Fig 1).

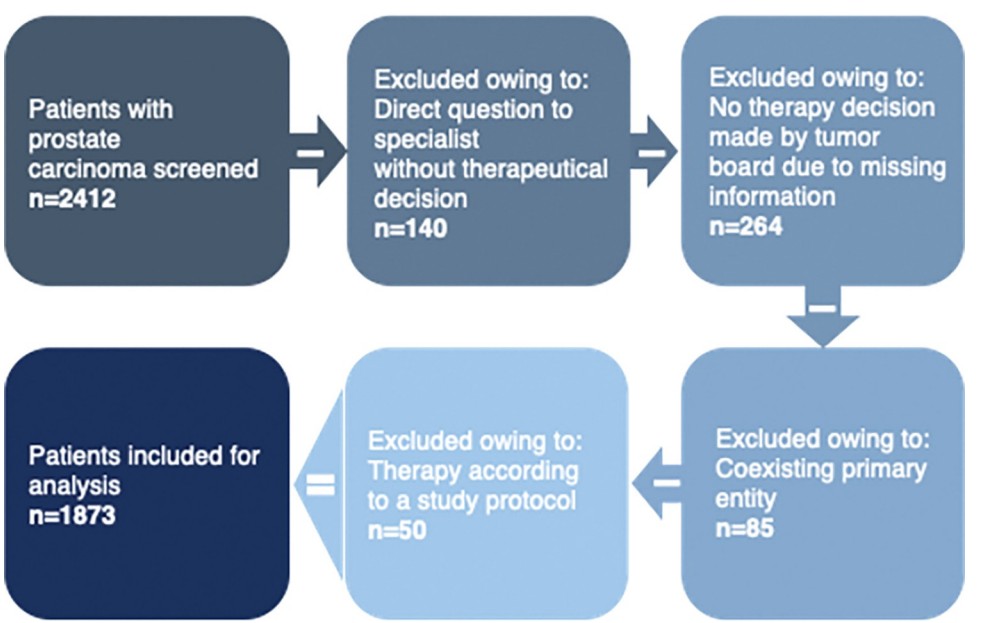

**Fig 1. Flow chart of patient case selection process.**

The real-world and digital treatment recommendations given for each tumor board question were compiled as response pairs and blinded to their origin. Subsequently, the response pairs were first examined for agreement and responses that were not obviously identical.

The similarities and deviations of each pair of answers were evaluated by independent uro-oncology specialists and reported according to previous publications on Watson for Oncology by IBM (Fig 2) [9–12,15]:

1. "concordant, recommended" if both recommendations of the DSS application and MTB were identical

2. At first evaluation non-concordant cases, that were reviewed by independent specialist and judged as correct alternative treatment option were considered as "concordant, for consideration"

3. Case pairs were classified as "non-concordant, not recommended" if one of the recommendations, either App or MTB, did not meet current best-clinical-practice treatment guidelines

4. Pairs of cases were "non-concordant, not available" if the DSS could not provide a treatment recommendation due to missing information that is needed by the APP to give a treatment recommendation

## Smartphone application

The content of EasyOncology was created by experienced clinical specialists from different cancer centers and oncology practices. Diagnostic and therapeutic recommendations follow the usual evidence-based guidelines of the professional societies, best clinical practice and the approval status of oncological therapeutics. The intuitive user interface and quality of the application led to a top 3 ranking in a worldwide comparison of 157 oncological applications in 2017 [16]. Certification as a class I medical device followed in July 2020.

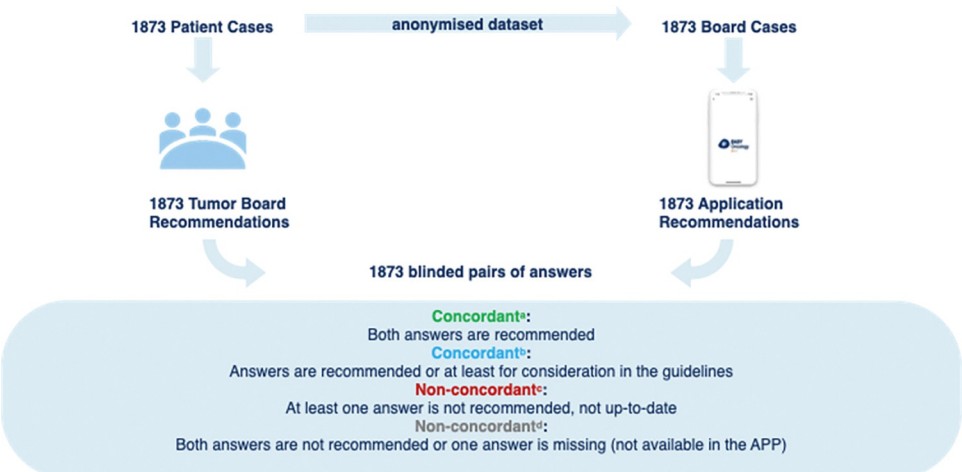

**Fig 2. Evaluation flow chart.** Testing results were categorized into 4 color-coded groups: green[a] represents "concordant treatment recommendation"; blue[b] represents "concordant, for consideration"; red[c] represents "non-concordant, not recommended" and grey[d] represents "non-concordant, not available" recommendations. In the second round of analysis, the mismatched pairs of responses were reviewed in detail in order to identify limitations in the query algorithm leading to non-concordancy and, subsequently, to improve the query. In summary, the evaluation method in this study involved comparing and analyzing the concordance rates between the tumor board recommendations of the urological multidisciplinary tumor board and the query results of the digital application "EasyOncology" and classifying the responses into different categories based on their agreement and compliance with best clinical practices.

Medical editors revised new content, discussed conflicting information with the authors, and assured EO updates in predetermined time intervals. Quality of decision algorithms was assured by implementing a constant comparison with real world decisions given in tumor boards.

As depicted in Fig 3 EO requests clinicopathologic patient data to generate treatment recommendations in a stepwise fashion. The number of clinicopathologic variables necessary to generate treatment recommendations depends on the complexity of the patient case.

For example in case of a "localized PC" only two clinicopathologic variables (i.e. risk group and histology) are required by EO to give a treatment recommendation. In more complex cases (i.e. nmCRCP) three clinicopathologic variables are requested by EO to provide a treatment recommendation.

## Data analysis and statistics

Descriptive statistics and data analysis were carried out using IBM´s statistics software SPSS version 25 and Microsoft Excel version 16. The patient characteristics age, cancer stage, risk stratification, Gleason Score, and PSA-level (prostate specific antigen) were documented. Descriptive statistics were depicted as number of percentages or mean ± standard deviation (SD). After assigning patients to the concordant or the non-concordant group, a chi-squared test was used to compare categorical variables and the Mann-Whitney U test was applied to compare ordinal variables between the groups. Multivariate logistic regression analysis was used to analyze the association between the concordance rate and clinicopathological data.

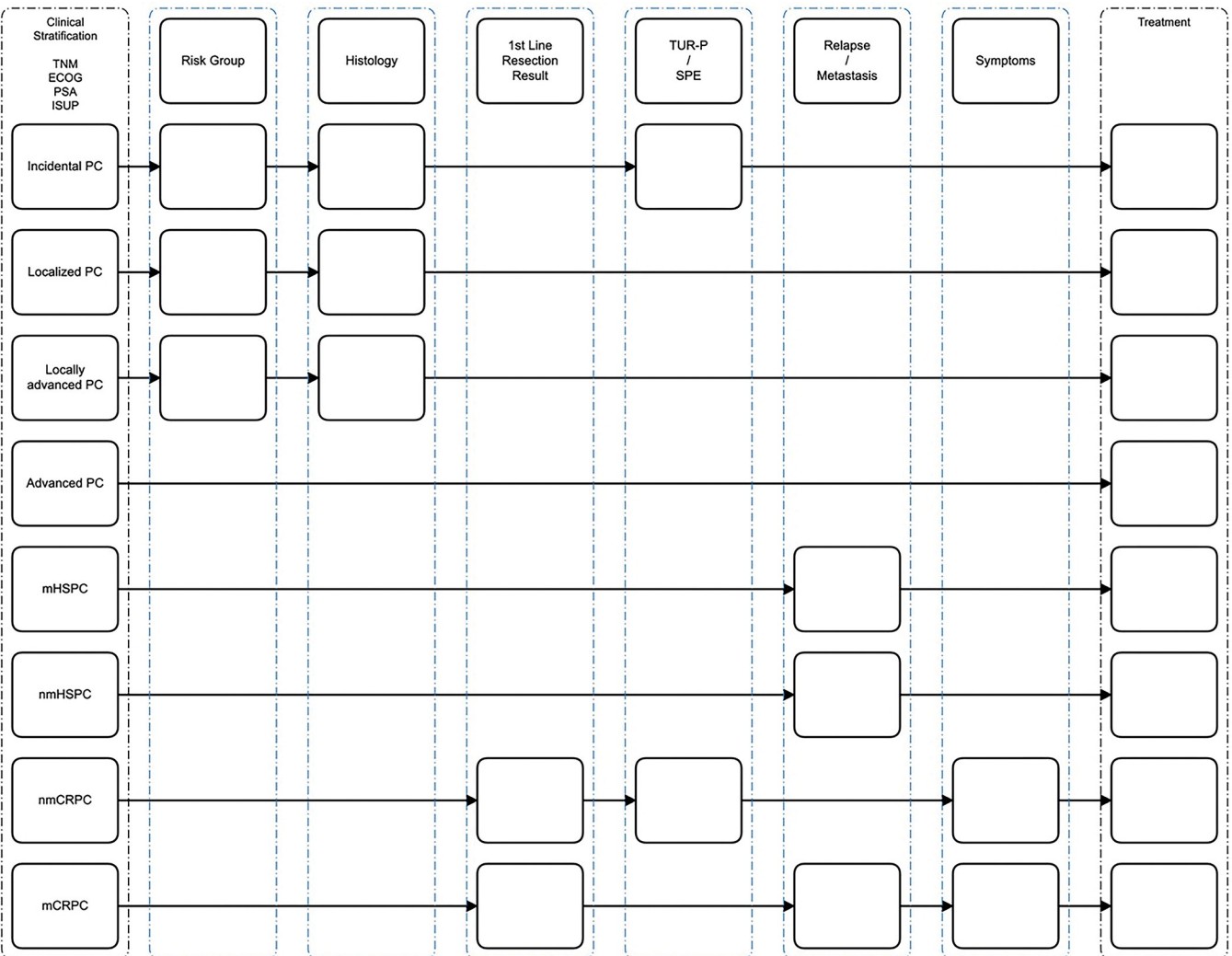

**Fig 3. Query algorithm.** The relevant information is requested by EO's query algorithm depending on the selected initial clinical status. Abbreviations: TNM: Classification of Malignant Tumors; ECOG: Eastern Cooperative Oncology Group; PSA: Prostate-Specific Antigen; ISUP: International Society of Urological Pathology; PC: prostate cancer; mHSPC: metastatic Hormone-Sensitive Prostate Cancer; nmHSPC: non-metastatic Hormone-Sensitive Prostate Cancer; nmCRPC: non-metastatic Castration-Resistant Prostate Cancer; mCRPC: metastatic Castration-Resistant Prostate Cancer; TUR-P: Transurethral Resection of the Prostate; SPE: Suprapubic enucleation.

Statistical significance was assumed if the p-value was $< 0.05$ for all statistical analysis. Graphics, charts and tables were generated using SPSS, Microsoft Excel and Power Point.

## Results

The mean age of all patients was 68 years. Regarding stage classification, 238 (13%) of cases were classified as stage I, 519 (28%) as stage II, 262 (14%) as stage III, and 848 (45%) as stage IV. Of the 776 cases with localized disease, 331 cases (43%) presented with good prognosis, 330 (42%) with intermediate prognosis, and 115 cases (15%) with poor prognosis according to D´Amico classification.

Cases were categorized according to clinical status as non-metastatic hormone-naïve and treatment-naïve prostate cancer (46%); metastatic hormone-naïve prostate cancer (11%); and castrate resistant prostate cancer (19%). A further subset included cases dealing with follow-

up; or treatment options after radical prostatectomy (RPE) with a histological R1 or pN1 situation; local or biochemical relapses; or questions regarding trans urethral or suprapubic resection of the prostate (TUR-P/SPE) after incidental detection of prostate cancer (24%).

Using multivariate analysis, the Gleason score and the prognostic Grade (ISUP) were significantly associated with concordance rate (p = .001 each). This analysis and other patient characteristics are summarized in Table 1.

The overall concordance rate between the actual treatments received by patients and cancer treatment recommendations given by EasyOncology for prostate cancer was 99%. Fig 4 shows the overall concordance rate.

As illustrated in Fig 5, stage specific concordance rates were 97.4% (stage I), 99.2% (stage II), 100% (stage III), and 99.2% (stage IV). Quality of concordance was independent of age, stage of disease and risk profile. The treatment concordance rates by age for < 50 years, 50–60 years, 60–70 years, 70–80 years, and ≥ 80 years were 100%, 99.0%, 99.6%, 99.0% and 97.0%, respectively. Patients with stage III cancer or who were <50 years old showed the highest concordance rates (100%).

Overall, non-concordant results were found in 17 cases (Fig 5). As requested by protocol, all non-concordant cases were reviewed by an independent uro-oncological specialist for exact sub-classification of non-matching cases.

After review, eight of these cases were classified as "non-concordant, not recommended"; nine cases as "non-concordant, not available".

Exemplary for a result that was rated "non-concordant, not recommended" to the disadvantage of the MTB was the case of a patient with newly diagnosed localized prostate cancer (UICC stage IIIA). Since the patient had a high PSA-level, the application recommended surgery or radiation, whereas MTB decided for active surveillance.

The correct recommendation of the APP according to guidelines was confirmed by the reviewer. Nevertheless, MTB decision followed the patient's request for non-intervention.

Another similar example of a "non-concordant, not recommended" case for independent review was a patient with localized prostate cancer (UICC stage I). The application recommended a therapeutic intervention in accordance with current guidelines, as two positive biopsies were documented in the board protocol [17,18]. Here again, individual decision (patient request) led to the MTB recommendation for active surveillance.

In the remaining six "non-concordant, not recommended" cases, the DSS recommended an active therapy based on the information that more than two biopsies were positive, indicating a higher-risk disease. MTB attending specialists realized that these biopsies were obtained only from one single tumor lesion, which is not fulfilling high-risk criteria, and therefore correctly dismissed the idea of an active therapy in favor of active surveillance.

Nine "non-concordant, unavailable" cases were identified as stage III neuroendocrine carcinomas, a histologic subtype that is not thematically considered by the APP and thus, no therapeutic recommendation was provided by DSS.

Using multivariate logistic regression analysis with independent variables age, PSA value and stage of disease (I/II vs. III/IV) no variables were found to be significantly associated with a decrease in the concordance rate. The results of the multivariate logistic regression are detailed in Table 2.

## Discussion

The rapid development of new, innovative oncological treatment options leads more than ever to the requirement of quality-assured therapeutic decisions [17]. In order to give optimal treatment recommendations, physicians usually follow guidelines of medical societies, inform

**Table 1. Baseline clinical characteristics.**

| Characteristic | Total, n = 1873 | Concordant group | Discordant Group | P Value |
|---|---|---|---|---|
| **Age, mean ± SD** | **68.3 ± 8.6** | **68.2 ± 8.6** | **71.1 ± 10.1** | **.168** |
| Disease stage (UICC), n (%) | | | | |
| I | 238 (12.7) | 232 (12.4) | 6 (0,3) | .140 |
| II | 519 (27.7) | 515 (27.5) | 4 (0,2) | |
| III | 262 (14.0) | 262 (14.0) | 0 (0.0) | |
| IV | 848 (45.3) | 841 (45.0) | 7 (0,4) | |
| N/A | 6 (0.3) | 6 (0.3) | 0 (0.0) | |
| PSA-level ng/ml, n(%) | | | | |
| ≤ 10 | 994 (53.1) | 982 (52.4) | 12 (0.6) | .399 |
| 10–20 | 393 (21.0) | 393 (21.0) | 0 (0.0) | |
| 20–50 | 213 (11.4) | 211 (11.3) | 2 (0,1) | |
| 50–100 | 120 (6.4) | 119 (6.4) | 1 (0,0) | |
| ≥ 1000 | 153 (8.2) | 151 (8.1) | 2 (0,1) | |
| Gleason Score, n(%) | | | | |
| 5 | 7 (0.4) | 7 (0.4) | 0 (0.0) | **.001** |
| 6 | 351 (18.7) | 344 (18.3) | 7 (0,4) | |
| 7a | 378 (20.2) | 374 (20.0) | 4 (0,2) | |
| 7b | 268 (14.3) | 267 (14.2) | 1 (0,0) | |
| 8 | 285 (15.2) | 284 (15.2) | 1 (0,0) | |
| 9 | 336 (17.7) | 336 (17.9) | 0 (0.0) | |
| 10 | 54 (2.9) | 54 (2.9) | 0 (0.0) | |
| N/A | 194 (10.4) | 190 (10.1) | 4 (0,2) | |
| ISUP prognostic grade, n(%) | | | | |
| I | 358 (19.1) | 351 (18.7) | 7 (0,4) | **.001** |
| II | 378 (20.2) | 374 (20.0) | 4 (0,2) | |
| III | 268 (14.3) | 267 (14.3) | 1 (0,0) | |
| IV | 285 (15.2) | 284 (15.1) | 1 (0,0) | |
| V | 390 (20.8) | 390 (20.8) | 0 (0.0) | |
| N/A | 194 (10.4) | 190 (10.1) | 4 (0,2) | |
| Clinical stage (D'Amico), n(%) | | | | |
| localized disease | 331 (17.7) | 324 (17.3) | 7 (0,4) | .086 |
| good prognosis | 330 (17.6) | 328 (17.5) | 2 (0,1) | |
| intermediate prognosis | 115 (6.1) | 114 (6.0) | 1 (0,0) | |
| poor prognosis | | | | |
| advanced disease | | | | |
| Clinical stratification | | | | |
| non-metastatic | 856 (45.9) | 846 (45.1) | 10 (0,5) | .694 |
| hormone-naive and | 203 (10.9) | 202 (10.7) | 1 (0,0) | |
| treatment-naïve cancer | 361 (19.4) | 357 (19.0) | 4 (0,2) | |
| metastatic hormone-naïve | 444 (23.8) | 442 (23.6) | 2 (0,1) | |
| prostate cancer | | | | |
| castrate resistant prostate | | | | |
| cancer | | | | |
| follow-up/therapy options | | | | |
| after RPE, local/ | | | | |
| biochemical relapse, TUR- | | | | |
| P/SPE with incidental | | | | |
| prostate cancer | | | | |

bold values indicate p < .005.

N/A info not given in board protocol

themselves through professional journals, participate in congresses, further their medical education and discuss cases in multidisciplinary tumor boards (MTB).

However, who can guarantee that all doctors working in oncology have the time and motivation to handle the information overload? How do they deal with this situation when even current guidelines of the medical associations sometimes fail to mention highly effective and

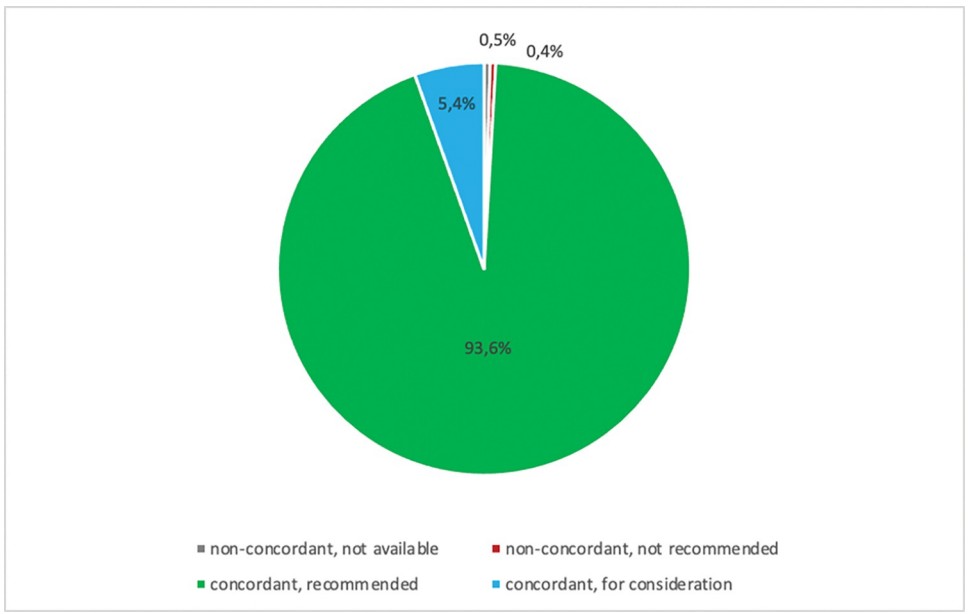

**Fig 4. Overall treatment recommendation concordance between a multidisciplinary tumor board and the application EasyOncology.**

newly approved therapeutics? Who ensures that the expertise of the doctors attending the MTB is actually given and that decisions are not (consciously or unconsciously) influenced by economic motives? Is there any evidence at all that tumor boards really improve the quality of oncology care [3,17–19]?

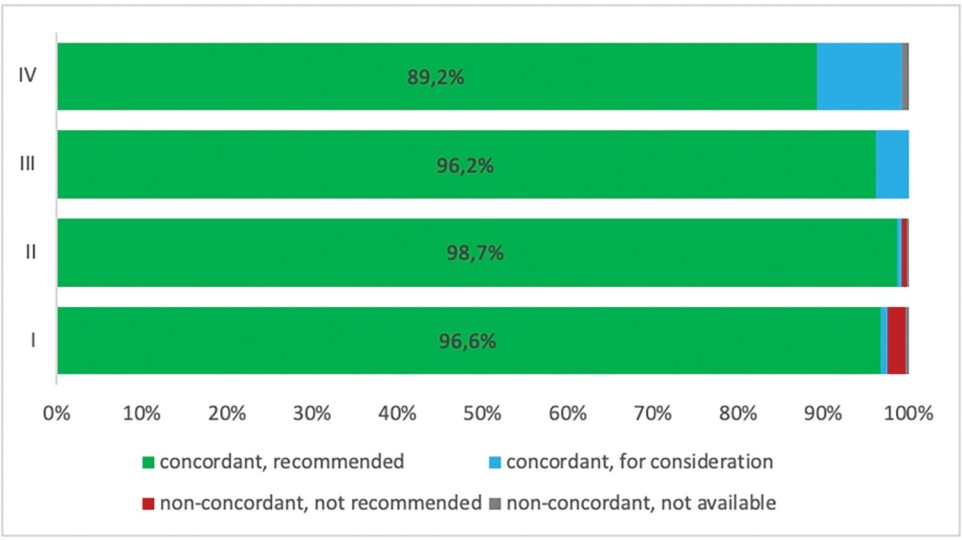

**Fig 5. Treatment concordance rates between a MTB and DSS according to prostate cancer tumor stage.** The queries of the APP on how many biopsy specimens were obtained, the patient's wish against any active therapy and the presence of a neuroendocrine tumor could thus be identified as systematic errors for divergent treatment recommendations. These valuable insights can be used to optimize the APP in order to increase the reliability of its recommendations.

**Table 2. Multivariate regression analyses of the concordance rate between EasyOncology and the multidisciplinary tumor board.**

| Variables | Multivariate Analysis | | |
|---|---|---|---|
| | OR | 95% CI | P value |
| Age | .955 | .898–1.015 | .141 |
| PSA (ng/ml) | 1.000 | 1.000–1.001 | .985 |
| Stage (≥3) | 2.210 | .832–5.870 | .112 |

AI-based systems seem to have the potential to support clinical decision making, as they have already impressively demonstrated their outstanding superiority in medical image processing and interpretation for different cancer entities [20–23]. Especially since AI-based systems seem perfectly suited to capture and correlate the immense amount of oncological knowledge, the results of all relevant clinical trials and all published case reports, and, based on this knowledge, to finally generate therapeutic recommendations. As obvious as this sounds, it is almost surprising that AI-based systems have yet failed to establish themselves in clinical oncological routine. So far, most attempts of AI-systems to reliably provide even standard therapeutic recommendations for first-line therapy have been disappointing. For example, Watson for Oncology, the leading application in this field, showed a concordance rate of only 73.6% compared to recommendations made by medical professionals for the first-line therapy of prostate cancer [16]. Watson for Oncology also obtained comparatively low concordance rates in other tumor entities, such as 12% in gastric cancer [13,24], 46.4% to 65.8% in colon cancer [10,11] or 77% in differentiated thyroid carcinoma [25] and others [9,26,27]. When considering the implementation of AI systems, the framework provided by the existing healthcare system must be carefully considered.

The effort required to implement systems such as Watson for Oncology in hospitals is enormous. The use of AI systems requires data protection-compliant interoperability between many hospital information systems and the required data must be completely accessible in predefined files.

In Germany in particular, data protection requirements of 16 federal states and the non-standardized norms for data processing and storage pose considerable challenges for developers of AI systems. In addition, clinical data continues to be frequently stored in paper files and it should not be forgotten that the exchange of diagnostic reports between clinics, pathologists, oncologists and practices is to this day commonly carried out by a fax machine.

Another point of criticism of AI systems is often that the decision-making process is not easy to understand and that one has to trust almost blindly in the correctness of the machine response. Furthermore, the validation efforts that are needed when using AI systems ties up considerable human resources since only medical experts can judge if the recommendations are correct.

Until these structural problems are solved, expert-curated solutions offer an alternative, as described in Nature Biotechnology in 2018 [14]. This approach was adopted by medical professionals using the DSS and is the basis of this research. In order to ensure the quality of the recommendations given by the application, a continuous comparison with tumor boards of certified cancer centers was implemented.

This comparably simple and resource-saving technical solution proved to be beneficial here, enabling the large number of tumor board recommendations to be effectively compared retrospectively.

For prostate carcinoma, our expert curated digital decision support system provides an optimal concordance rate with the therapeutic recommendations of a university tumor board.

Yet, the very high concordance rate of our system is probably not surprising, since we evaluated predominantly first-line cases, for which guidelines generally apply. Of course, the degree of complexity increases with each tumor recurrence and additional concomitant diseases.

However, this is exactly the specification of our work, which aims to reduce the workload of tumor boards by providing digitalized answers to non-complex routine cases.

Despite the fact that this approach achieves better results than other methods published so far, further limitations have to be taken into account.

It should be stated as a limiting factor of our work that a high concordance rate is easier to achieve when therapeutic strategies do not show a significant change over a longer period of time, as given during the time period studied, from 2014 to 2018 [27].

The increasing dynamics of diagnostic and therapeutic options in the treatment of prostate cancer thus leads to significantly more frequent and shorter testing intervals of the application, which has been certified as a medical device in the meantime, and to continuous adaptation to best-clinical-practice.

By continuously comparing digital and analog recommendations, systemic deficits that lead to deviations usually become quickly apparent, thus enabling the prompt adaptation of the query logic to the dynamic development of therapeutic options.

Second limiting bias, a 100% concordance rate is of little value if the recommendation quality of the reference board is not validated. This leads to the necessity of establishing decision networks in order to generate a recommendation basis that is as reliable as possible and provides the basis for the required safety of recommendations. Therefore, cross-validation of different urological cancer centers is ongoing in order to eliminate the single-center bias. Another limitation is certainly that the reference recommendations are based on the German S3 guidelines and thus the methods and results cannot be transferred to other countries uncritically.

The main goal of this development is to provide reliable recommendations for standard cases in advance of the tumor board conference with the aim of allowing more time for the discussion of complex cases. It is not in the developers' interest to achieve 100% agreement between tumor board responses and digital recommendations, as no automated system will be able to consider all the complex clinical circumstances. The agreement of 100% achieved in our analysis in stage III indicates only the simpler decision for active therapy in this risk constellation compared with the early stages of prostate cancer.

Rather, a trusted DSS must be able to reliably identify complex clinical constellations, which should then be discussed by experts attending the tumor board. Particularly in the case of complex diseases, medical expertise is irreplaceable and must remain so. However, expertise requires time, and this time should not be spent discussing universally accepted standard procedures.

## Conclusion

In summary, the study evaluated a decision support system (DSS) for first-line therapy of prostate cancer by comparing its recommendations with those made by a multidisciplinary tumor board (MTB) at a university cancer center. The study found a high level of concordance between the DSS-generated recommendations and those of the MTB, indicating a high level of reliability. Continuous analysis of mismatched cases ensures early adjustment of DSS recommendations to account for changes in best clinical practice. Overall, our results suggest that EO is a promising tool to assist clinicians in providing reliable treatment recommendations for prostate cancer patients.

### Perspective options

It is almost paradoxical in evidence-based driven oncology that the actually relevant quality of individual therapeutic decisions is virtually unknown.

The use of intelligent software could ensure the quality of treatment on a case-by-case basis and thus serve as an instrument for quality assurance that can be transparently accessed and compare the quality of oncological care provided by hospitals and medical practices.

Based on the smartphone application used in this work for recommendation matching, we developed an interface that enables the necessary inputs in the decision process of tumor boards. Easily integrated into any system, this validated and reliable application could unburden tumor boards from standard cases, thereby allowing more time for discussion of complex cases.

## Author Contributions

**Conceptualization:** Yasemin Ural, Thomas Elter.

**Data curation:** Yasemin Ural, Yasemin Yilmaz.

**Formal analysis:** Yasemin Ural.

**Funding acquisition:** Yasemin Ural, Thomas Elter.

**Investigation:** Yasemin Ural, Thomas Elter.

**Methodology:** Yasemin Ural, Thomas Elter, Rabi Raj Datta, Robert Kleinert.

**Project administration:** Thomas Elter, Axel Heidenreich.

**Resources:** Michael Hallek, Axel Heidenreich, David Pfister.

**Software:** Thomas Elter.

**Supervision:** Michael Hallek, Axel Heidenreich, David Pfister.

**Validation:** David Pfister.

**Visualization:** Yasemin Ural, Thomas Elter, Yasemin Yilmaz.

**Writing – original draft:** Yasemin Ural, Thomas Elter.

**Writing – review & editing:** Axel Heidenreich, David Pfister.

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
