## [Editor Report · Decision Letter 0]

31 May 2022

PDIG-D-22-00125

Validation and implementation of a mobile app decision support system for quality assurance of tumor boards. Analyzing the concordance rates for prostate cancer from a multidisciplinary tumor board of a University Cancer Center

PLOS Digital Health

Dear Dr. Ural,

Thank you for submitting your manuscript to PLOS Digital Health. After careful consideration, we feel that it has merit but does not fully meet PLOS Digital Health's publication criteria as it currently stands. Therefore, we invite you to submit a revised version of the manuscript that addresses the points raised during the review process.

The major weakness of the paper lies in its methodology section. This could be improved by providing more details on

1. Framework or theory involved in the methods

2. Rationale for statistical analysis including appropriate statistical analyses/methods

3. Rationale for the study design and why a design like RCT wasnt chosen

4. Inadequate details on the EasyOncology application and this could be improved in the main text

5. Funding source of this study- Needs to be explicit. How was this study funded? 

We look forward to receiving your revised manuscript.

Kind regards,

Padmanesan Narasimhan, MBBS MPH PhD

Section Editor

PLOS Digital Health

Journal Requirements:

1. Please provide separate figure files in .tif or .eps format only and remove any figures embedded in your manuscript file. Please also ensure that all files are under our size limit of 10MB.

For more information about how to convert your figure files please see our guidelines: https://journals.plos.org/digitalhealth/s/figures

---

## [Decision Letter · Decision Letter 1]

14 Nov 2022

PDIG-D-22-00125R1

Validation and implementation of a mobile app decision support system for quality assurance of tumor boards. Analyzing the concordance rates for prostate cancer from a multidisciplinary tumor board of a University Cancer Center

PLOS Digital Health

Dear Dr. Ural,

Thank you for submitting your manuscript to PLOS Digital Health. Our reviewers have got back to us and suggested minor revisions. Therefore, we invite you to submit a revised version of the manuscript that addresses the points raised during the review process.

Please submit your revised manuscript within 30 days Dec 14 2022 11:59PM. If you will need more time than this to complete your revisions, please reply to this message or contact the journal office at digitalhealth@plos.org. Please include the following items when submitting your revised manuscript:

We look forward to receiving your revised manuscript.

Kind regards,

Padmanesan Narasimhan, MBBS MPH PhD

Section Editor

PLOS Digital Health

Journal Requirements:

Additional Editor Comments (if provided):

Reviewers' comments:

Reviewer's Responses to Questions

**Comments to the Author**

1. If the authors have adequately addressed your comments raised in a previous round of review and you feel that this manuscript is now acceptable for publication, you may indicate that here to bypass the “Comments to the Author” section, enter your conflict of interest statement in the “Confidential to Editor” section, and submit your "Accept" recommendation.

Reviewer #1: (No Response)

Reviewer #2: All comments have been addressed

2. Does this manuscript meet PLOS Digital Health’s publication criteria? Is the manuscript technically sound, and do the data support the conclusions? The manuscript must describe methodologically and ethically rigorous research with conclusions that are appropriately drawn based on the data presented.

Reviewer #1: (No Response)

Reviewer #2: Yes

3. Has the statistical analysis been performed appropriately and rigorously?

Reviewer #1: (No Response)

Reviewer #2: N/A

4. Have the authors made all data underlying the findings in their manuscript fully available (please refer to the Data Availability Statement at the start of the manuscript PDF file)?

Reviewer #1: (No Response)

Reviewer #2: Yes

5. Is the manuscript presented in an intelligible fashion and written in standard English?

Reviewer #1: (No Response)

Reviewer #2: Yes

6. Review Comments to the Author

Reviewer #1: I thank the authors for this manuscript draft, and especially the great work they are doing in developing EasyOncology DSS for oncology. I believe their approach in using DSS, in combination with / controlled by the expert clinicians is exactly the right method to approach as complex and heterogenous indication area as oncology with various cancers and subpopulations is. Authors’ high level objective in this work is in replacing the valuable time of MTB in assessing trivial cases with DSS based automation is also correct approach when moving stepwise from AI assisted humans to more fully automated decision making. 

I do have some comments and proposals on the latest article revision. I believe some of them are more critical before this manuscript is ready to be published. 

GENERAL COMMENTS

• You correctly mention in the abstract and discussion, that cross-validation over multiple clinics is required to further increase decision quality and avoid potential clinic bias. This is exactly as it should be, however as the EO application and advice by DSS evaluated in this manuscript are based(?) on PCa treatment practises specifically in Germany, I am left to consider whether your DSS and cross-validation is then applicable to treatment practises in Germany only. At least EO is available in German only. I can only assume based on my personal experience that SoC varies a lot in different countries, especially so in the earlier stages on PCa. I believe this should be mentioned as a limitation, at least in the discussion, or if the CV is done over several countries, perhaps this should be then mentioned to tackle this challenge?

• Authors use both MTD and MTB abbreviations for multidisciplinary tumor board. Please unify. 

• In general, should the figure titles be also within the figures too? The quality of figure images is also rather poor.

• Check the font for different header levels in the manuscript for ease readability.

• Overall, the references in the text to table 1, figure 3 and figure 4 do not seem to be perfect in relation to their aimed location within the text.

• Use of references, square brackets vary through the manuscript a bit, at least in discussion. Sometimes after the period, sometimes before. Please unify.

Table 1:

• Some values contain size of N/A population, in some cases such as clinical stage & stratification the size of not mentioned. 

• What is the “a” in superscript in the footnote?

Figure 1: 

• please check the language of figure title

Figure 4: 

• please elaborate the title 

ABSTRACT

• In the author summary you write “Unfortunately, current digital tools that have been developed to support clinicians on the process of decision-making, have failed to provide treatment recommendations with sufficient accuracy” – now this is a rather bold and should perhaps be softened by using “having difficulties” or similar, instead of “failing”. 

• You also speak here about newly diagnosed patients here but do not mention anything about this in the study design and patients? Was this an inclusion criteria for retrospective analysis, and if not, are the results applicable for patients with longer patient journey? There seems to something in the discussion about this but I believe it should belong to design and patients, as well.

INTRODUCTION

• In the end of introduction I would suggest a small change: “The aim of this clinical research is to implement the aforementioned technology for validation and quality assurance of a urological tumor board in NAME OF HOSPITAL”.

MATERIAL AND METHODS

Smartphone application

• I do believe the authors that EO has been adequately developed, and credible, but the manuscript itself is still in my opinion failing to answer to original review critique #4: what is it that EO does? The reader would need to download the application (not available in the country where I live). What are the possible treatment recommendations (input => OUTPUT)? I can understand what the input is, but I don’t understand what the output from DSS is here? It would increase the contextual understanding of these results a lot.

• What is the medical device classification of EO?

Study design and patients

• How is the second round of analysis relevant for these results or this manuscript?

DATA ANALYSIS AND RESULTS 

• MS Excel version, if it really has been used for data analysis?

DISCUSSION

• 2nd paragraph: I would change the questions into statements that are then backed by references.

• You are writing that “all attempts of AI-systems to reliably provide even standard therapeutic recommendations for first-line therapy have been disappointing”. I wouldn’t use such a strong statement with one example. 

• Finally, I would touch shortly the use case vs reliability here – is it better that system over-recommends treatments as what are the consequences when someone who needs to be treated won’t get it? On the other hand, safety issues are waiting if over-recommending treatments.

Reviewer #2: (No Response)

7. PLOS authors have the option to publish the peer review history of their article (what does this mean?). If published, this will include your full peer review and any attached files.

**Do you want your identity to be public for this peer review?** For information about this choice, including consent withdrawal, please see our Privacy Policy. 

Reviewer #1: No

Reviewer #2: Yes: Yacine HADJIAT

---

## [Decision Letter · Decision Letter 2]

24 Feb 2023

PDIG-D-22-00125R2

Validation and implementation of a mobile app decision support system for quality assurance of tumor boards. Analyzing the concordance rates for prostate cancer from a multidisciplinary tumor board of a University Cancer Center

PLOS Digital Health

Dear Dr. Ural,

Thank you for submitting your manuscript to PLOS Digital Health. After careful consideration, we feel that it has merit but does not fully meet PLOS Digital Health's publication criteria as it currently stands. Therefore, we invite you to submit a revised version of the manuscript that addresses the points raised during the review process.

Please submit your revised manuscript within 30 days Mar 26 2023 11:59PM. If you will need more time than this to complete your revisions, please reply to this message or contact the journal office at digitalhealth@plos.org. Please include the following items when submitting your revised manuscript:

We look forward to receiving your revised manuscript.

Kind regards,

Haleh Ayatollahi

Section Editor

PLOS Digital Health

Journal Requirements:

2. Please send a completed 'Competing Interests' statement, including any COIs declared by your co-authors. If you have no competing interests to declare, please state "The authors have declared that no competing interests exist". Otherwise please declare all competing interests beginning with the statement "I have read the journal's policy and the authors of this manuscript have the following competing interests:"

3. Please ensure that Funding Information and Financial Disclosure Statement are matched.

4. In the Funding Information you indicated that no funding was received. Please revise the Funding Information field to reflect funding received.

Additional Editor Comments (if provided):

The topic of the manuscript is interesting and it is well-written. Please consider addressing the following issues in your manuscript as well.

1- The title seems too long. Please make it shorter. The authors can remove the second part of the title starting with “Analyzing…”.

2- Please choose the keywords based on the MeSH terms, as well.

3- Please follow the journal instructions to provide a concise unstructured abstract.

4- In the introduction, although the authors reviewed the existing literature, they need to explain the gap in the existing knowledge, too.

5- In the methods section, please explain the study design and participants first and then explain the design of the smart phone application.

6- In page 5, the authors noted “The content of EasyOncology was developed by experienced specialists…”. It is important to elaborate on this part and make it clear how and using which methodology the content was developed.

7- In the methods section, please add the inclusion and exclusion criteria for the patients.

8- The readers may need to know more about the technical aspects of the CDSS. Please add some figures of the system and provide an example (flow chart) to show how the system worked. Moreover, please make sure that the figures are visible in the submitted file.

9- The evaluation methods need to be explained in detail. The authors could also evaluate the specificity, sensitivity, accuracy, etc.

10- In Table 1, some figures are the same and their percentages are different and vice versa. Please re-check Table 1.

11- The possible reasons for reaching concordance rates of (100%) need to be explained in the discussion section.

12- We need to see a conclusion section after the discussion.

Reviewers' comments:

Reviewer's Responses to Questions

**Comments to the Author**

1. If the authors have adequately addressed your comments raised in a previous round of review and you feel that this manuscript is now acceptable for publication, you may indicate that here to bypass the “Comments to the Author” section, enter your conflict of interest statement in the “Confidential to Editor” section, and submit your "Accept" recommendation.

Reviewer #1: All comments have been addressed

2. Does this manuscript meet PLOS Digital Health’s publication criteria? Is the manuscript technically sound, and do the data support the conclusions? The manuscript must describe methodologically and ethically rigorous research with conclusions that are appropriately drawn based on the data presented.

Reviewer #1: Yes

3. Has the statistical analysis been performed appropriately and rigorously?

Reviewer #1: Yes

4. Have the authors made all data underlying the findings in their manuscript fully available (please refer to the Data Availability Statement at the start of the manuscript PDF file)?

Reviewer #1: Yes

5. Is the manuscript presented in an intelligible fashion and written in standard English?

Reviewer #1: Yes

6. Review Comments to the Author

Reviewer #1: All the comments were addressed in a satisfying manner.

7. PLOS authors have the option to publish the peer review history of their article (what does this mean?). If published, this will include your full peer review and any attached files.

**Do you want your identity to be public for this peer review?** For information about this choice, including consent withdrawal, please see our Privacy Policy. 

Reviewer #1: Yes: Sammeli Liikkanen

---

## [Editor Report · Decision Letter 3]

27 Apr 2023

Validation and implementation of a mobile app decision support system for prostate cancer to improve quality of tumor boards

PDIG-D-22-00125R3

Dear None Ural,

We are pleased to inform you that your manuscript 'Validation and implementation of a mobile app decision support system for prostate cancer to improve quality of tumor boards' has been provisionally accepted for publication in PLOS Digital Health.

Best regards,

Haleh Ayatollahi

Section Editor

PLOS Digital Health